# Network Analysis of Survey Data to Identify Non-Homogeneous Teacher Self-Efficacy Development in Using Formative Assessment Strategies

**Jesper Bruun** *,[†] and **Robert Harry Evans** [†]

Department of Science Education, University of Copenhagen, 1350 Copenhagen K, Denmark; evans@ind.ku.dk
* Correspondence: jbruun@ind.ku.dk; Tel.: +45-2627-4203
† These authors contributed equally to this work.

**Abstract:** In a European project about formative assessment, Local Working Groups (LWGs) from six participating countries made use of a format for teacher-researcher collaboration. The activities in each LWG involved discussions and reflections about implementation of four assessment formats. A key aim was close collaboration between teachers and researchers to develop teachers' formative assessment practices, which were partially evidenced with changes in attributes of self-efficacy. The research question was: to what extent do working with formative assessment strategies in collaboration with researchers and other teachers differentially affect individual self-efficacy beliefs of practicing teachers across different educational contexts? A 12-item teacher questionnaire, with items selected from a commonly used international instrument for science teaching self-efficacy, was distributed to the participating teachers before and after their work in the LWGs. A novel method of analysis using networks where participants from different LWGs were linked based on the similarities of their answers, revealed differences between empirically identified groups and larger super groups of participants. These analyses showed, for example, that one group of teachers perceived themselves to have knowledge about using formative assessment but did not have the skills to use it effectively. It is suggested that future research and development projects may use this new methodology to pinpoint groups, which seem to respond differently to interventions and modify guidance or instruction accordingly.

**Keywords:** formative assessment; in-service teacher development; self-efficacy; network analysis

## 1. Introduction

### 1.1. The Need for Monitoring Changes in Formative Assessment Practices

Formative assessment is widely seen as an important way of helping students learn science (e.g., [1]). In this study, providing formative assessment was defined as when science teachers make student- and criterion-based judgments about learning progress based on data from student activities, provide guidance for students' next steps in learning and suggestions as how to take these steps ([2], pp. 55–59). However, many teachers experience a gap between theoretical understandings of formative assessment and how to implement formative assessment methods in practice [3]. Helping teachers and researchers work together to bridge the gap between theory and practice may lead to a higher probability that teachers use formative assessment to further student learning.

In the context of this study, science teachers from six European countries worked with researchers to apply four different methods for formative assessment in their own teaching practice. The four formative assessment methods were quite broad to allow teachers from different contexts to

adapt them in their teaching. The methods were written teacher feedback, on-the-fly assessment, classroom dialogue, and peer-feedback [4]. After applying each method, teachers and researchers discussed and reflected on the utility of formative assessment in the classroom. This study used attributes of self-efficacy as indicators of the potential for teacher development with formative assessment methods, to change teacher practice.

The reasons teachers may have for being able to implement formative assessment methods in their own practice are likely quite diverse and complex, as are teachers' responses to working together with researchers. That is, the dynamics of how teachers respond to a given "treatment" likely varies a lot. Often this diversity and complexity can be hidden in average scores and changes in pre intervention to post intervention designs [5]. In fact, many studies rely on basic pre intervention to post intervention changes to overall self-efficacy without going into details of these changes, see for example [6–8]. However, we argue that there is a lack of depth and understanding in the literature pertaining to the diversity of challenges and possibilities for teacher self-efficacy in using formative assessment. This study addresses this problem by using network analysis of teacher responses to known items for measuring attributes of self-efficacy to empirically examine outcomes for teachers who worked with researchers to develop their assessment practices. It shows how network analysis can help make detailed, data-driven analyses of attributes of self-efficacy and of changes in these attributes from pre intervention to post intervention.

## 1.2. Raising Teacher Self-Efficacy through Peer and Researcher Collaboration

Self-efficacy is a capacity belief that can be an indicator of a teacher's confidence that they can successfully perform specific teaching acts. Bandura [9] wrote that self-efficacy beliefs '…contribute significantly to human motivation and attainments'. These values were relevant to this study, since we were interested in facilitating teacher use of various forms of formative assessment during their science teaching. Both teachers' motivation to use formative assessment and their ability to do so were related to their relevant self-efficacies. We measured variables which contribute to self-efficacies before and after several years of iterative uses in order to assess changes that would indicate differences in their judgement about their capacity to use formative assessment in teaching.

Self-efficacy beliefs are malleable and constantly changing as on-going experiences are incorporated into current levels of self-efficacy. Consequently, measured changes in self-efficacy attributes can be useful in following changes in beliefs over time. Bandura [10] proposed four ways in which self-efficacy beliefs can be altered: 'enactive mastery experiences', 'vicarious experiences', 'verbal persuasion' and 'affective states'. We actively used all four methods throughout the research project to influence participant teacher's self-efficacies. Most important was our facilitation of opportunities for 'enactive mastery experiences' which occurred when teachers got feedback about their use of formative assessment through repeated attempts after which they could adjust their methods to improve their usage. Teachers also adjusted their self-efficacies for formative assessment through 'vicarious experiences' when hearing about the attempts of their peers in using these methods, they could infer about their own ability to apply them. We facilitated opportunities for these 'vicarious experiences' through periodic formal Local Working Group (LWG) meetings where experiences with formative assessment were shared as well as through informal exchanges among local school colleagues. These same meetings were also opportunities for us as researchers to use 'verbal persuasion' to help LWG teachers accommodate their experiences and accurately judge their degree of successful use of the methods. During all of these scenarios, the 'affective states' of the teachers, whether physiological anxiety over a challenging new method or an attitude about testing the methods, played a continuous background role in changing self-efficacies.

The LWGs in each country (Czech Republic, Denmark, Finland, France, Germany, Switzerland) met several times a year with project researchers during trials of the four strategies for formative assessment. Teachers were introduced to each strategy at these meetings and at subsequent meetings they reported on and discussed their perceptions about affordances and challenges with one another as

well as with local researchers. Minutes of these meetings in each of the six participating countries were shared with project leaders to monitor the extent to which formative strategies were implemented. These minutes were also used to ensure that the general objective for LWG members to have opportunities to gain confidence in their use of formative assessment methods was achieved.

By measuring these changes, we aimed to evaluate the effects of introducing the four formative assessment methods into teaching repertoires. We hypothesized that by actively using Bandura's [10] four methods for self-efficacy change, we could influence teacher motivation and success with formative assessment. This is based on the work of other researchers who identified the sources of change in these capacity beliefs. Bautista [11] studied the effectiveness of 'enactive mastery' and 'vicarious' experiences. Results suggested that enactive mastery, cognitive pedagogical mastery as well as vicarious experiences of symbolic modelling and cognitive self-modelling were the major sources of positive influence on the teacher self-efficacies, a result consistent with [10]. In a subsequent study, Bautista & Boone [8] found enactive mastery of cognitive learning, virtual modelling and cognitive self-modelling and emotional arousal were the primary sources of increase in pre-service elementary teacher's perceived self-efficacy beliefs.

Typical frameworks for investigating self-efficacy are well established (see e.g., [12]), but in this paper, we aim at adding to typical analyses of empirical evidence by looking at specific attributes which contribute to self-efficacy and employing a new methodological approach, network analysis. Rather than the general overall changes in self-efficacy measured by a cohesive instrument we examined some of the specific component attributes of self-efficacy with individual belief items. We argue that network analysis allows us to use empirical data to find various groups with similar self-efficacy attributes rather than looking for differences in a priori categories, such as gender [13]. Thus, the paper focuses on the implementation of that approach and on new interpretations which emerge in that process.

*1.3. Choosing Network Analysis as an Analytical Tool for Questionnaire Data*

In this article, we use network analysis as an analytical tool, and provide a novel method for analysing questionnaire data using network analysis. To warrant our choice of network analysis as our analytical tool, this section first gives a short review of network analysis as a methodological tool in science education with the purpose of showing how network analysis is designed to highlight relational patterns in data. For example, such analysis is useful in finding and showing clusters or groups of entities, the internal structure of clusters, and between cluster structures. Since other methodological tools for finding cluster structure in questionnaire data exist, we will also briefly review other clustering methods and argue why network analysis may hold some advantages over these methods.

Network analysis has been used in science education research to map social connections between students, teachers, and other stakeholders, and also how these connections change over time (e.g., [14–16]). For this reason, network analysis is often equated with social network analysis. However, this study does not rely on social networks nor on the theoretical frameworks developed in the Social Sciences to understand these [17]. Rather, this study uses networks where connections resemble similarity [18], which we call similarity networks, to capture patterns and nuances of teacher responses, regardless of Local Working Group affiliation.

It is crucial for any interpretation of networks to make clear, which characteristics of individuals are represented and which characteristics connections signify. In network analysis, nodes represent aspects of entities of interest while links (usually drawn as lines) represent connections between nodes [19] (see Figure 1). In this study, nodes represent aspects of teachers, such as Local Working Group, science subjects, teaching experience, and self-efficacy attributes. Links in this study represent to which degree teachers answered similarly to a self-efficacy questionnaire (see Construction, grouping, and analysis of similarity networks for an overview and Supplemental File S1. Calculating similarity for details).

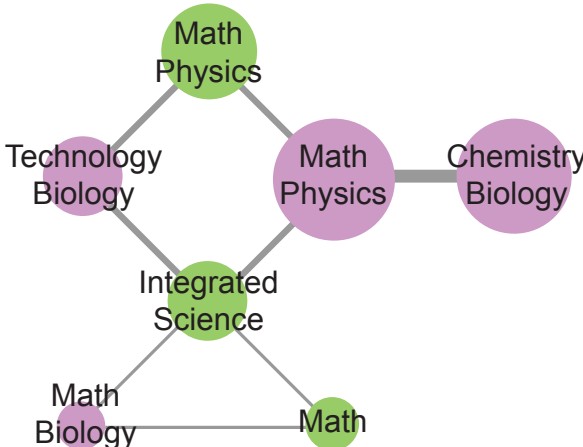

**Figure 1.** Similarity network constructed example. Nodes (circles) represent teachers, and links (lines) represent similarity of teacher responses. Thicker lines represent more similar responses. Graphically, nodes can show different aspects of teachers, such as self-efficacy scores or demographics. For example, node size could represent mean self-efficacy attribute scores, color could represent demographic information (e.g., country or gender), and text could represent subject taught.

This way of constructing networks allows a detailed and nuanced analysis of teacher response patterns and thus helps understanding of the potential implications for formative assessment practices through measures of self-efficacy. Based on these patterns, groups of similarly answering teachers, without regard to the LWG to which they belong, can be identified and characterised. This means that the method developed in this study depends on teachers' actual responses rather than on any a priori assumptions about behavioural differences inferred from background variables. While network representations of data have been used previously to find group-structure in data [20,21], other methods also exist. Most methods include calculating the similarity of objects of study and using a computer algorithm to determine which are grouped together [22]. The network methodology we propose here is in some respects similar to existing methods in that the objective is to find cluster structure in the data. This can also be done, for example, via K-means or hierarchical cluster analysis [23], factor analysis [24] and mixture models [25]. We argue though that network analysis retains more information about the structure of connections [26]. Like the above mentioned alternatives, network analysis provide information about who is grouped together, but also about the structure within each clusters (or as we will call them, groups) and how clusters relate to one another in terms of similarity.

This in turn affords detailed analyses, as will be illustrated here. In our study, science teachers from across Europe tried out and discussed similar formative assessment approaches in different educational contexts. Their development was monitored through answering a questionnaire on attributes of self-efficacy. Network analysis helped us find groups of similarly answering teachers. This can be seen as a data-driven stratification of the empirical material that will help us find differential responses of practicing teachers to participation in the research project on formative assessment.

## 2. Research Question

The research question to be answered by using the analytical methodology proposed here is: To what extent does working with formative assessment strategies in collaboration with researchers and other teachers differentially affect individual self-efficacy attributes of practicing teachers across different educational contexts?

With this question, we emphasise that our approach is data driven. This is represented by the word "differentially." We do not assume that teachers from any predefined group will behave in the same manner. In contrast, we intend for our analytical methodology to be able to deal with teacher similarities and differences across any a priori groups.

### 3. Methods

*3.1. Participating Teachers and Controls*

The cohort consisted of 101 science and mathematics teachers from Local Working Groups in six participating European countries [27]. Teachers were locally selected based on teaching experience, subject expertise, and availability. The questionnaire was distributed to the 101 teachers in these Local Working Groups. 64 teachers answered the questionnaire's pre intervention and post intervention administrations. In addition to the self-efficacy attributes instrument described below, we collected a number of background variables: years of experience with teaching (four categories: less than 4 years, 5–10 years, 11–20 years, and more than 20 years), teaching subject, and Local Working Group. 13% had less than 4 years of experience, 25% 5–10, 41% had 11–20, and 22% had more than 20 years of experience. Of the respondent teachers, 52% taught math, 36% biology, 30% physics, 30% chemistry, 8% technology, and 16% taught integrated science, meaning courses where the focus is on scientific processes and not on content.

*3.2. Local Working Group Intervention*

In each of the participating countries a Local Working Group consisting of country researchers and teachers was formed. The LWGs were structured around what was called trials in the project. These were half-year periods during which teachers tested particular teaching units and methods for formative assessment in their own classrooms. The project had three such trials. For each of the trials in all country sites, Local Working Groups met 3 or 4 times with one another, and with project leaders, to plan teaching and use of formative assessment and to discuss the results and experiences. During implementations, LWG teachers tried the assessment methods and reflected both individually and as groups on the results of their trials.

The teacher trials with formative assessment methods were designed to provide opportunities for 'enactive mastery experience' with the four formative assessment methods since they were tried multiple times with intervals between for reflection and feedback. The project engaged experienced teachers whose self-reflections after repeated lesson trials were likely to have influenced their self-efficacies, positively and negatively, for each of the methods they used. In addition, since they met with peers in their LWGs before, during and after trials, the opportunities for 'vicarious experience' through the influences from the group were frequent. Concomitantly, there were opportunities for influential members of the LWGs as well as project leaders to affect teacher self-efficacies through social 'verbal persuasion' at meetings where the processes and results of the trials were discussed. Repeated trials over the three trials gave opportunities for adjustments in 'physiological and affective states'. In addition to possibly increasing the success of introducing special formative assessment methods, these intentional opportunities for self-efficacy change provided a method of gauging changes in teacher self-efficacy attributes relevant to their use of formative assessment.

Each LWG was situated in a particular country. However, any one LWG in this study cannot be said to be representative of the country in which it was situated. Thus, we do not interpret the results of our analyses in light of country.

*3.3. Self-Efficacy Instrument*

A pre intervention and post intervention teacher questionnaire was administered to all project teachers from the six LWGs. Amongst a total of 73 questions, it contained 12 items whose aim was to assess self-efficacy attributes of teachers low in familiarity with various formative assessment methods. These 12 items were adapted from a commonly used international instrument for overall science teaching self-efficacy [12,28]. This instrument, STEBI-B, contains 23 Likert-type scale items with two constructs consistent with self-efficacy theory, both a person's belief about their capacity to perform a future task as well as their judgement about outcome expectations within the context where their performance will occur [10].

The 12 self-efficacy attribute items in this project's questionnaire include both of these constructs (see Figure 2). For example, Question 38 is only about self-efficacious beliefs (confidence in using formative assessment in future teaching), whereas Question 43 includes an outcome expectation due to context (inadequate student backgrounds). Unlike the STEBI-B questionnaire which assesses overall science teaching self-efficacy, each of these 12 items only assess one attribute of self-efficacy relative to formative assessment, independently of the others. These 12 items do not represent a standardized instrument to measure self-efficacy and have been analysed at the item level elsewhere [27]. Each was given in a 1 to 5 level Likert-type format with a sixth option of 'no response'. Using network analysis, this article investigates how to use these individual question responses to find patterns of similarity within participants.

---

Q38. I will continually find better ways to teach using formative assessment.

Q39. Even if I try very hard, it will be difficult for me to integrate formative assessment

into my teaching. (R)

Q40. I know the steps necessary to teach effectively using formative assessment.

Q41. I will not be very effective in monitoring student work when I teach using formative

assessment. (R)

Q42. My teaching will not be very effective when using formative assessment. (R)

Q43. The inadequacy of students' background can be overcome by using formative

assessment. (*)

Q44. When a low-achieving student progresses, it is usually due to formative assessment

given by the teacher. (*)

Q45. I understand formative assessment well enough to be effective using it.

Q46. Increased effort of the teacher in using formative assessment produces little change

in some students' achievement in inquiry-based competencies. (R*)

Q47. When using formative assessment, I will find it difficult to explain subject content to

students. (R)

Q48. I will typically be able to answer students' questions when using formative

assessment.

Q49. I wonder if I will have the necessary skills to use formative assessment. (R)

---

**Figure 2.** The twelve separate attributes of self-efficacy assessment items from a longer questionnaire with their original item numbers. Scoring reversed (R) for Questions 39, 41, 42, 46, 47 and 49. Items with a * assess 'outcome expectations' whereas all of the others specifically assess 'self-efficacy belief attributes'. (Adapted from [12,28]).

### 3.4. Construction, Grouping, and Analysis of Similarity Networks

We have argued that a network analytical approach is useful for finding groups of teachers that answer similarly within groups but differently across groups. However, any theoretical knowledge produced by a research study is necessarily shaped by the methodology that underpins it. Thus, we include a full technical description of our proposed novel methodology in the Supplemental File (Sections 1–4, and 6). This section outlines the methodology and details the rationale behind it.

The methodology developed for this study relies on calculating similarity between respondents. The question was how to define similarity in a meaningful way. In some studies (e.g., [29]), similarity has been established using principal component analysis (PCA) or factor scores. In such a model, two people are similar, if the distance between them in a mathematical 'component' or 'factor' space is small. Such a procedure requires that the raw data is amenable to principal component or factor analysis FA). Since the number of questions in this study's questionnaire is small as is the number of participants, we would not expect either PCA or FA to produce meaningful results. Instead, this study follows Lin [30] and uses overlapping responses as a basis for similarity. An overlapping response for two teachers for a particular question will add to the similarity between those two teachers, whereas different answers will not add to their similarity. The similarity measure takes the frequency of responses into account; an overlapping response chosen by most teachers will not contribute as much to two teachers' similarity as less frequently chosen responses. For example, say that two teachers respond by choosing Likert-type option '2' on question 38, and most other teachers have also responded '2'. This overlapping response will contribute less to their similarity than if these two teachers respond "5" and most other teachers responded '2'. This means that the measure is sensitive to nuanced ways of understanding teacher self-efficacy in the context of the study.

Teacher similarity scores on the self-efficacy questions were calculated between each responding participating teacher. These scores were then used to create a similarity network between respondents. This was done by considering each respondent as a node and similarity scores as links. See Figure S1 in the Supplemental File for details and Figure 1 for a constructed example.

Most teachers will have some degree of overlap, simply because with five possible responses, they sometimes select the same responses to some of the questions. This results in a highly connected—or dense—network, which is difficult to use for differentiating between teachers' self-efficacy responses. In network science, a solution to the problem of highly connected networks is to remove some of the connections. In this study, we adopt locally adaptive network sparsification (LANS) [31]. The method compares each of a teacher's similarity connections to the rest of that teacher's similarity connections. If the connection is numerically larger than or equal to a predefined fraction of that teacher's other connections (for example, 0.95), the connection is said to be significant (for example, at the 5% level) and is kept. Otherwise it is said to be insignificant and is removed. Thus, the method mirrors the standard measures of significance. Since the method trawls through each teacher's connection, and each connection has two ends, a connection can be significant for the teacher at one end but not for the teacher at the other end. In this study, a similarity connection is kept if it is significant for at least one teacher. The result is a network where connections resemble overlaps on responses to 12 self-efficacy attribute items (see Figure 3a), which are significant (for example, at the 5% level) from at least one teacher.

The next step in the methodology was to differentiate between different teachers' self-efficacy attribute responses. The strategy was to find groups of teachers that respond similarly to particular self-efficacy questions. Network analysis offer a variety of methods for finding groups of nodes in networks. Collectively, these methods are called community detection methods (see e.g., [32]), although we stress that we cannot expect groups found in a similarity network to resemble a socially based community. This study relies on the Infomap method for community detection [33], see Supplemental File S4. Community detection for finding groups for details. The Infomap method produces groups of nodes that represent similar teacher responses. The groups are rooted in each individual teacher's responses in relation to other individual responses and allows for empirical

determination of different self-efficacy profiles. It is standard in community detection to measure the quality of the grouping by applying two quantitative measures. One is called 'modularity' and it measures the number of connections within groups relative to what could be expected randomly. The other measures the 'robustness' of the grouping by applying the community detection method many times and measuring the stability of groups. These two measures are described in more detail in the Supplemental File S4.

*3.5. Analysis of Groups*

Having constructed and found groups in a similarity network depicting self-efficacy attribute responses, this study analysed the groups in different ways.

Attributes of teachers in each group were reported and analysed from the perspective of over-representation. For example, teachers from a particular LWG or with a particular subject may be over-represented in a particular self-efficacy group. To quantify over-representation, we use the Z-score of a previously developed measure for over-representation [16]. As is standard for Z-scores, Z > 1.96 indicates significant over-representation. See Supplemental File S4 for details. The post intervention responses for each group were used to characterise the detailed response patterns for each group. This characterisation is in part the answer to the research question of this study; it is used to differentiate groups of teachers from each other.

Third, as illustrative examples of detailed differential self-efficacy attribute changes, a detailed analysis was made for selected questions for each group linking pre intervention responses with post intervention responses. Here we calculated the per group frequency differences for each Likert-type option for each question. This part of the analysis helps us investigate the detailed development of self-efficacy attributes for teachers in each group.

Statistical analyses were done using *t*-tests (when necessary assumptions could be met), effect sizes (Glass's Δ for different variances or Cohen's *d* with near equal variances), distribution entropies ($H = -\sum p_i \log_2 p_i$, see Supplemental Material), standard Z-scores, Kruskal-Wallis tests as non-parametric analysis of variance on frequency distributions, and Nemenyi's test for post hoc analyses of Kruskal-Wallis tests.

Calculations and network analyses were made using R statistical environment [34] using packages PMCMRplus [35], rcompanion [36], effsize [37], gplots [38], and igraph [39]. Visualisations were done using Gephi [40] and Mapequation.org [33].

## 4. Results

For the participants in the study, the methodology resulted in a pre intervention network ($N = 64$, $L = 72$) of teacher responses before participating and a post intervention network ($N = 64$, $L = 106$) of teacher responses after participating in the study. Both networks could individually be stably grouped, but groups were not stable between the two networks. This indicates that teachers changed their responses, but also that the network analysis did not identify particular groups of teachers that tightly followed each other in changing their specific responses to the questionnaire. Since this study focused on results of the intervention further network analyses focused on the post intervention network.

The community detection algorithm revealed 12 groups in the post intervention network, which were labelled 'Post Group 1' and so forth. A 'modularity' score of 0.58 indicated that overall the groups were distinct to a high degree [41]. However, some groups were very small ($N = 3$ for Post Group 12, for example), which made a detailed analyses frequency of answers intractable. To overcome this difficulty, we used both the network and the network map shown in Figure 3a,b to visually identify groupings of groups—'super groups'. The depiction of the network in Figure 3a was made using a force-based algorithm in Gephi. We used this algorithm numerous times and found the same structure of the layout every time, except that the layout was sometimes mirrored or rotated. Thus, for example, the green nodes of Post Group 2 would always be near the red nodes of Post Group 3 and the orange nodes of Post Group 1 in the same spatial configuration as shown in Figure 3a. We also

used this feature of the layout algorithm in our identification of super groups. However, since this was based on a visual inspection, we took a number of steps to ensure the validity and reproducibility of the super groups. Having identified candidate super groups, we checked the quality of the new grouping by confirming that the average internal similarity between teachers in super groups was larger than the external similarities—the average similarity between teachers in a super group and all teachers outside of that super group. The next paragraph explains the visual analyses, while the details of the quantitative analyses are given in the Supplemental File S6 Finding and quantitatively characterising the optimal super group solution.

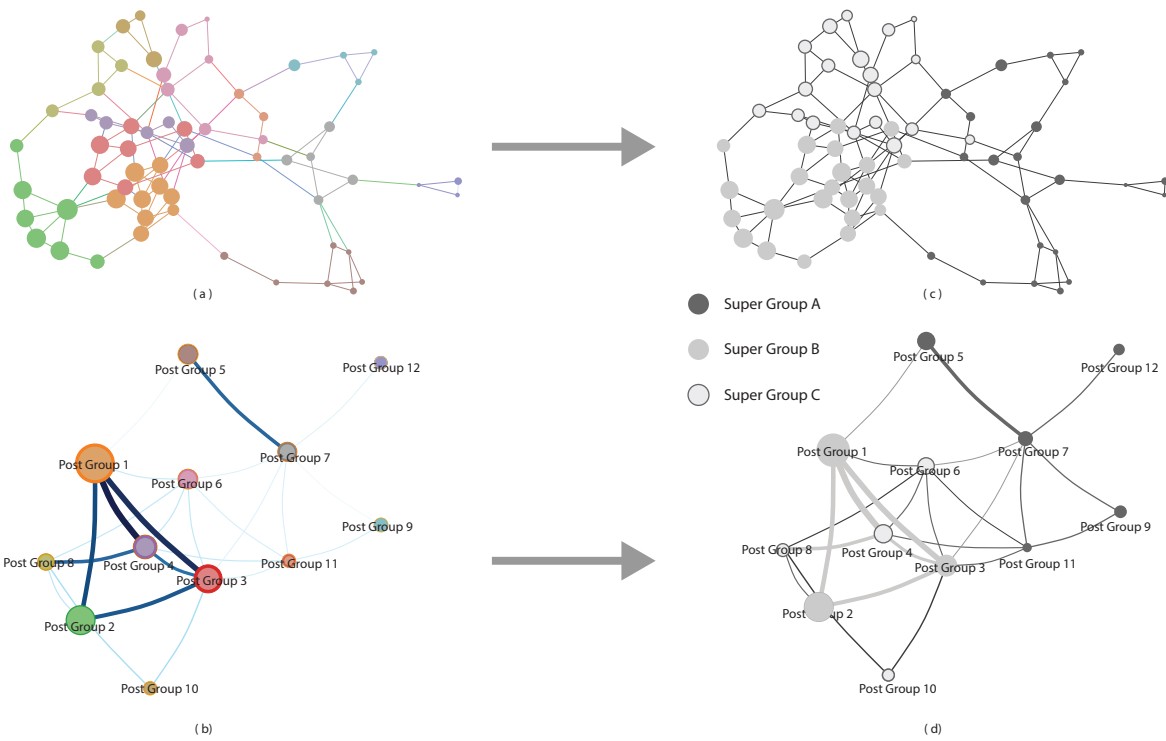

**Figure 3.** (**a**) The similarity network shows how each participant is connected via their post intervention self-efficacy attributes responses. Each group has its own color (**b**) The corresponding map of the similarity network shows similarity groups and how they are connected. The colors match, so that e.g., the orange nodes in (**a**) are collected in the orange Post Group 1 in (**b**). (**c**) shows the similarity post intervention network colored with the super group scheme. (**d**) shows corresponding map.

Based on a visual inspection of the network and network map as depicted in Figure 3a,b, we initially merged Post Groups 1–12 into Super Groups A, B, and C. Super Group A consisted of Post Groups 5, 7, 9, and 12, Super Group B of Post Groups 1–4, and Super Group C of Post Groups, 6, 8, 10, and 11. This initial grouping was made by investigating shared links in the network map (Figure 3c) and spatial placement in the layout of the network (Figure 3a). Since it was done by visual inspection, we made slight permutations of Super Groups to see if they would result in a better-quality solution. We used the Modularity, $Q$, as our measure of quality. Thus, we searched for the solution with three Super Groups that would maximise modularity. However, we also wanted to embed the Post Groups found empirically and robustly by Infomap in Super Groups. As such, our first rule for changing Super Groups was that whole Post Groups could switch between Super Groups. Our second rule was that Post Groups could only switch to a Super Group to which they were adjacent in the network map. The third rule, our decision rule, was that if a switch increased $Q$, the new solution was accepted, if not, it was rejected. If a new solution was accepted, the adjacency of all Post Groups to Super Groups were re-evaluated. The end result was Super Group A consisting of Post Groups 5, 7, 9, 11, and 12 (22 teachers); Super Group B consisting of Post Groups 1, 2, and 3 (23 teachers); and Super Group C

consisting of Post Groups 4, 6, 8, and 10 (19 teachers). Figure 3c,d show Super Groups A, B, and C. The modularity of the new grouping was $Q = 0.515$, which is lower than the optimal solution but still indicative of a strong group structure [41]. The analyses of similarities yielded an average of internal similarities of 0.51 ($SD = 0.09$), which was significantly higher ($t(85.59) = 3.895, p < 0.0005, \Delta = 0.99$, 95% CI = [0.56, 1.43]) than the average external similarities of 0.45 ($SD = 0.05$).

The measure for over-representation revealed that some LWGs were over-represented in some Super Groups ($Z = 9.85, p < 10^{-5}$). Table 1 shows the distribution of LWG per super group. Thus, Super Group A consists mainly of teachers from LWG 2 and all of LWG 2's teachers are in Super Group A. Furthermore, all four teachers in LWG 5 and most teachers from LWG 6 were in Super Group B.

**Table 1.** The distribution of Local Working Group members in Super Groups A, B, and C. Notice that Super Group C primarily consists of members of LWG 2

|  | Super Group A | Super Group B | Super Group C | *Total* |
|---|---|---|---|---|
| LWG 1 | 1 | 6 | 6 | 13 |
| LWG 2 | 14 | 0 | 0 | 14 |
| LWG 3 | 2 | 3 | 2 | 7 |
| LWG 4 | 5 | 1 | 4 | 10 |
| LWG 5 | 0 | 4 | 0 | 4 |
| LWG 6 | 0 | 9 | 7 | 16 |
| *Total* | 22 | 23 | 19 | 64 |

We also tested for over-representation in terms of subject, years of teaching, and gender. Two other results emerged. First, we found that ten participating teachers who taught integrated science were placed in Super Groups A (six teachers) and B (four teachers). Closer inspection revealed that the six integrated science teachers in Super Group A belonged to LWG2 while the four integrated science teachers in Super Group B were from LWG5 and LWG6. Since these were the LWGs, which were over-represented in Super Groups A and B respectively, we believe that this result stems from that association rather than being an effect of teaching integrated science. We found no over-representation for other subjects. We found no evidence of over-representation in terms of years of teaching.

Next, we examined general response commonalities by looking at the frequency distributions for responses for each question for each of the three super groups. We did this for both pre intervention and post intervention responses. For their five Likert-type level selections, we treated a '1' as a low self-efficacious response while '5' was high. We maintained the not answered (NA) category in the results because it contains relevant information about the super groups. For four of the twelve questionnaire items we reversed the scores due to negatively worded self-efficacy questions (Q39, Q41, Q42, Q46, Q47 and Q49). These reversals were made before graphing and are therefore included in the graphs of Figures 4 and 5. Higher scores for every question mean higher self-efficacious answers. For clarity here, we have 360 reworded the question statements in the graphs for the six questions with reversed scores by slightly 361 rewording the questions so that a high self-efficacious answer yields a high self-efficacy score. (See the original questions in Figure 2 for comparison).

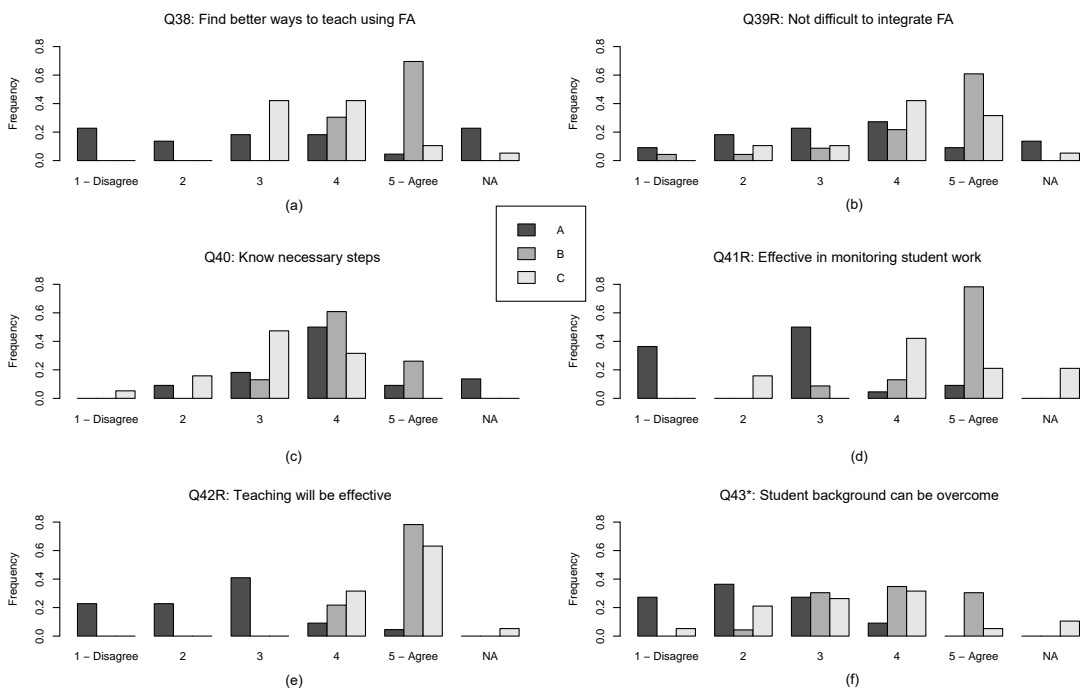

**Figure 4.** Post question frequency distributions for Super Groups A, B, and C, questions 38 to 43. R means that coding was in reverse, and * marks an outcomes expectation question.

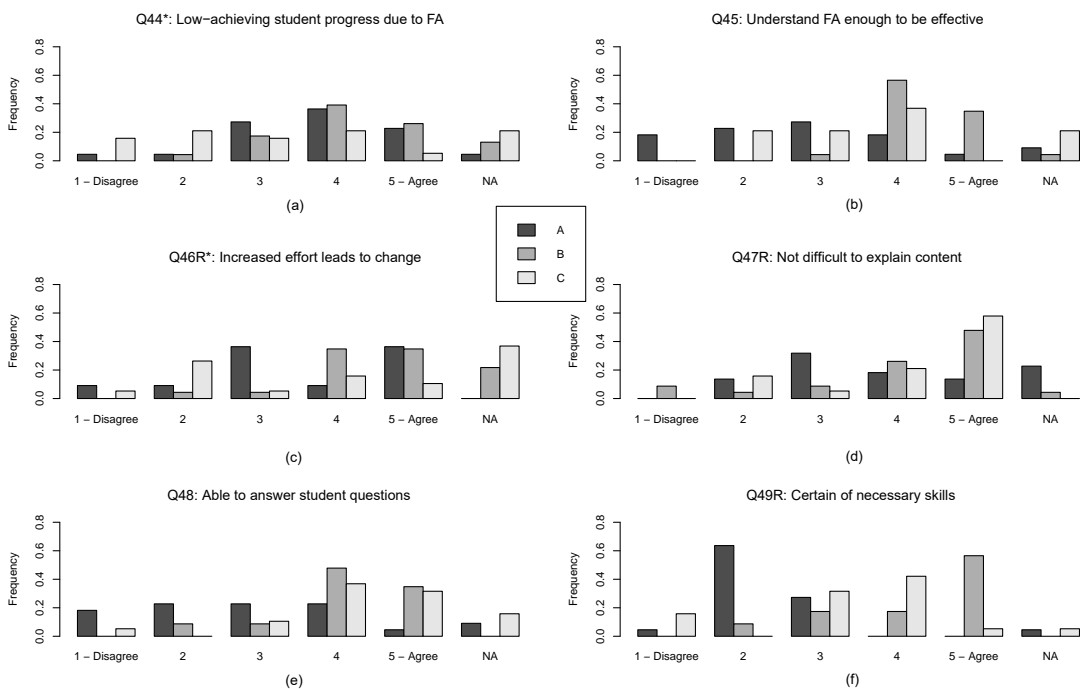

**Figure 5.** Post question frequency distributions for Super Groups A, B, and C. Questions 44 to 49. R means that coding was in reverse, and * marks an outcomes expectation question.

This closer question-by-question look at the twelve items clarifies the general network maps from Figure 3c,d. Overall, Super Group A's black bars for each item (see Figures 4 and 5) show a tendency toward less self-efficacious responses while the shaded bars of Super Groups B and C represent more self-efficacious responses with the medium grey bars of Super Group B trending the highest. Furthermore, a visual inspection indicates that Super Group A and C tend have more disparate answers

than Super Group B on self-efficacy attribute questions. For the outcome expectations questions (Q43, Q44, and Q46R), Super Group C tend to have more disparate answers. This visual observation needed to be quantified to make sure that it was not a visual artefact. This was done by calculating the entropy [42] of each distribution of responses to each question for each super group. See the Supplemental File S5 Technical results for details. In this case, the entropy tells us whether responses are restricted to few categories (low entropy) or more broadly distributed (high entropy, see e.g., [43,44]). For example, in Q38, Super Group A uses the full range of possibilities (high entropy), while Super Groups B and C primarily makes use two each (lower entropy). In contrast, in Q43, Super Group A and B each make use of primarily three categories (medium entropy), while Super Group C has a broader distribution. However, there are important additions to this broad characterisation, so next we characterise each super group in terms of their answers to specific combinations of questions. The point is to show that using this study's network methodology adds important nuance to how teacher self-efficacy attributes were differentially affected by participating in the project.

Super Group A indicates to a large extent that they know the necessary steps to teach effectively using formative assessment, since their answers are centred on 4 in Q40. The Kruskal-Wallis test revealed that there were significant differences ($\chi^2(2) = 16.3, p < 0.0005$) and the subsequent post-hoc test revealed that there were significant differences between Super Group B and C ($\chi^2 = 16.3, p < 0.0005$), but not between A and B ($\chi^2 = 2.81, p > 0.24$), and marginally between A and C ($\chi^2 = 5.10, p < 0.1$). Thus, Super Group A and B can be said to agree on the question of having the necessary knowledge.

While such an agreement is not found on question Q49R on having the have the necessary skills to provide formative assessment. Again, a Kruskal-Wallis test revealed significant differences ($\chi^2(2) = 28.4, p < 10^{-6}$). Here, there are significant differences between Super Group A and B ($\chi^2 = 28.3, p < 10^{-6}$), between A and C ($\chi^2 = 7.36, p < 0.05$), and marginally between B and C ($\chi^2 = 5.47, p < 0.1$). The Supplemental material provides detailed calculations for all question frequency distributions. 63% of Super Group A were coded with a 2 on Q49R, giving them the lowest entropy of the three groups on that question. The agreement of Super Group A on Q49R contrasts their answers to questions that may be seen to pertain to different formative assessment skills. The group is diverse on whether they perceive it as difficult to integrate formative assessment (Q39R), whether they will be effective in monitoring student work (Q41R, almost dichotomous between 1 and 3), whether their teaching with formative assessment will be effective (Q42R and Q45), and whether they will be able to explain content (Q47R) or able to answer student questions (Q48) using formative assessment. Thus, although the problem for this group seems to be that they do not believe that they have the necessary skills, it is different skills they miss. Super Group A shows somewhat less diversity with regards to outcome expectations. They generally do not seem to believe that the inadequacy of student background can be overcome by using formative assessment, but seem to agree that if a low-achieving student should progress, it will be because of the teachers' use of formative assessment (Q44).

This conundrum reflects a common divergence between teacher preparation and classroom action. For self-efficacy, we believe the explanation lies within the theoretical basis for self-efficacy, The work of Bandura [10] shows that beliefs that one can perform a task (such as providing formative assessment) are constantly mediated by the context of the performance. So, the self-efficacy attributes we measured were composed of 'capacity beliefs' and 'outcome expectations', meaning that even with strong teacher beliefs in a capacity to use formative assessment, a given teaching situation may make it difficult or impossible to implement. For example, the participants in this study learned the affordances of peer feedback in their teaching. However, when they implemented it, they sometimes found that their students were not skilled at providing useful peer feedback to one another and so the teachers either became reluctant to use it or they actively taught their students how to provide good feedback [45]. Their negative 'outcome expectation' based on novice student feedback was sometimes overcome

through student practice at giving feedback, thereby allowing the participant's high efficacy belief about peer feedback to be realized.

Super Group B can be contrasted with Super Group A in the sense that they both choose higher ranking categories (relevant $\chi^2$- and *p*-values for all questions are given in the Supplementary File) and also show less variety (lower entropy) on questions that pertain to formative assessment skill. This difference can be confirmed by returning to the network representation. Super Group B appears as a more tightly knit group in Figure 3c, and this is confirmed numerically in that the internal similarity of the group is higher than for the two other groups (see Supplemental File, Table S2). There is some variation in their answers to questions pertaining to outcome expectations (Q43, Q44, and Q46R). Interestingly, some chose not to answer Q44 (13%) and Q46 (21%). This may be due to the slight uncertainty of the effect of formative assessment in general even if they believe that they are very capable of providing formative assessment.

Super Group C represents a middle ground between Super Group A and B. The responses in this super group are typically marginally different from one super group while significantly different from the other. Interestingly, Super Group C has higher diversity (entropy) than the two other super groups with regards to outcome expectations (Q43, Q44, Q46R), knowing the necessary steps to teach effectively using formative assessment (Q40), and having the necessary skills to use formative assessment (Q49R). In this sense, Super Group C seems to represent a group of teachers for which knowledge, skills and even the general applicability of formative assessment is uncertain.

To investigate how Super Groups differentiate with respect to change in self-efficacy attributes, we compared pre intervention with post intervention frequency distributions, focusing on differences in change for Super Groups. Figure 6 compares pre intervention and post intervention assessment attribute choices and shows the differences, or shift, for each Likert-type option for each question. So taking graph for Q40 for example, the positive value of 0.21 for Likert-type option '3' for Super Group C is because 0.16 of Super Group C members chose '3' at pre intervention while 0.47 of Super Group C members chose this option at the post, signifying an positive shift in the number of respondents from this group choosing this option. Similarly, negative values signify a decrease.

In Figure 6, we notice apparent shifts towards more consensus for Super Groups on Q40 (know necessary steps). Super Group A's responses seem shifted towards option 4 (indicated by the positive value), as are Super Group B's. Super Group C's responses seem shifted towards option 3. To quantify these apparent shifts, we compared them with all 216 shifts ($M = 0, SD = 0.13$, see Supporting File for details) by calculating the Z-score, $Z = (x - M)/SD$. Compared to all shift-values, the shift in option '4' on Q40 for Super Group A was marginally significant ($Z = 1.73, p < 0.1$), for Super Group B on option '4', the shift was significant ($Z = 2.64, p < 0.01$), for Super Group C on option '3' the shift was also significant ($Z = 2.40, p < 0.05$). Furthermore, the negative shift for Super Group B on option '3' was marginally significant ($Z = 1.65, p < 0.1$). No other shifts in the distribution were significant. Thus, we can say that a positive shift occurred for each Super Group towards either a medium or a medium high level of agreement with Q40s statement, but we cannot claim that this resulted from any one negative shift on other options. Rather it seems that each Super Group's distribution narrowed towards medium or medium high levels of agreement with knowing the necessary steps to teach effectively using formative assessment. For Q49, only one shift was significant; Super Group A had a significantly positive shift on option '2' ($Z = 3.11, p < 0.005$). No other shifts were significant for this question. Thus, while Super Group B and C did not change their position on Q49, Super Group A shifted towards not being certain of having the necessary skills to use formative assessment. We have included pre intervention to post intervention graphs for all questions in the Supplemental Material.

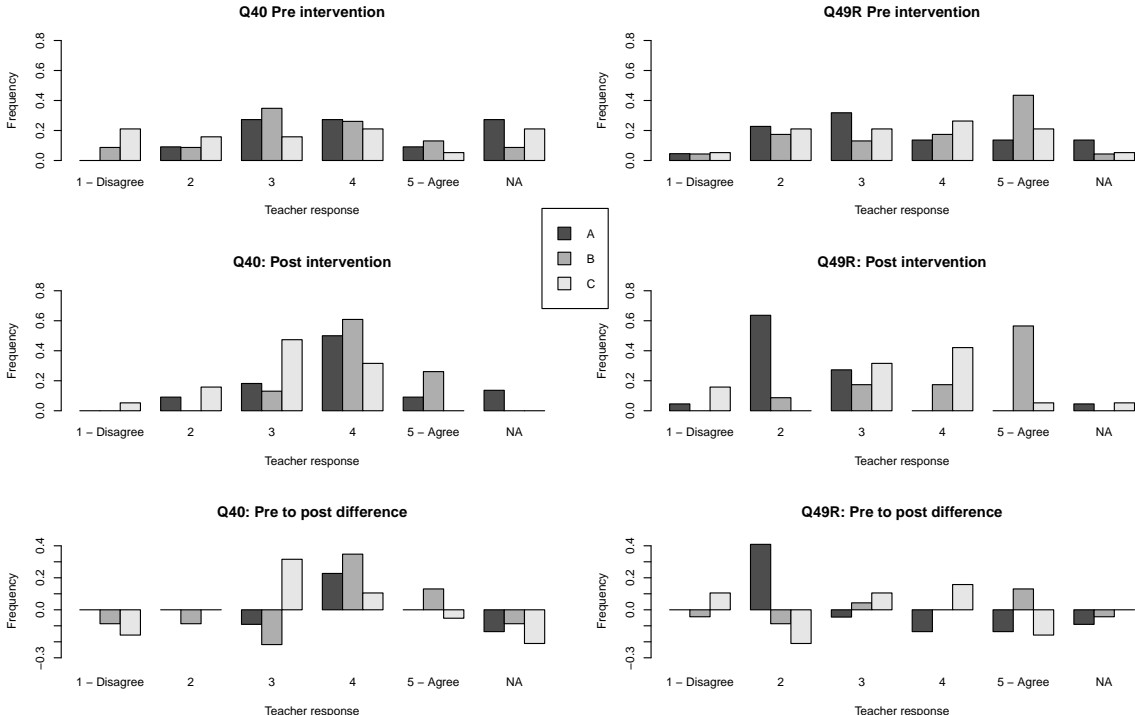

**Figure 6.** Pre intervention and post intervention frequency distributions and differences in these for Super Groups A, B, and C for questions 40 (I know the steps necessary to teach effectively using formative assessment) and 49 (I am certain that I have the necessary skills to use formative assessment). R means that coding was in reverse.

As a different illustrative example, we compared Super Group shifts on Questions Q42 and Q46. Figure 7 shows that Super Group B and C shifted their selection towards agreeing with the the self-efficacy attribute statement that their teaching would be very effective using formative assessment. The shifts towards option '5' were significant for Super Group B ($Z = 1.98$, $p < 0.05$), and for Super Group C ($Z = 2.80$, $p < 0.01$). Super Group C has some uncertainty as to whether an increased effort in using formative assessment will produce change in some students' achievement (options '2' and 'NA' are prevalent in Figure 7f). Super Group B, on the other hand, seems to be more certain that increased effort in using formative assessment may produce change in student achievement. This may be a subtle signifier that the intervention supported both groups in their beliefs that they are able to use formative assessment, but did not convince teachers in Super Group C that this would be beneficial to all students. Super Group B may have gone into the project with the belief that formative assessment would be beneficial to students and did not change that belief.

These examples of detailed analyses of Q40, Q49, Q42, and Q46 serve as illustrations of the differential responses and patterns of movement for each Super Group. While we believe that these two illustrative findings could potentially be important, any interpretation would be speculative, and a richer analyses involving e.g., interview data would be needed to substantiate this picture.

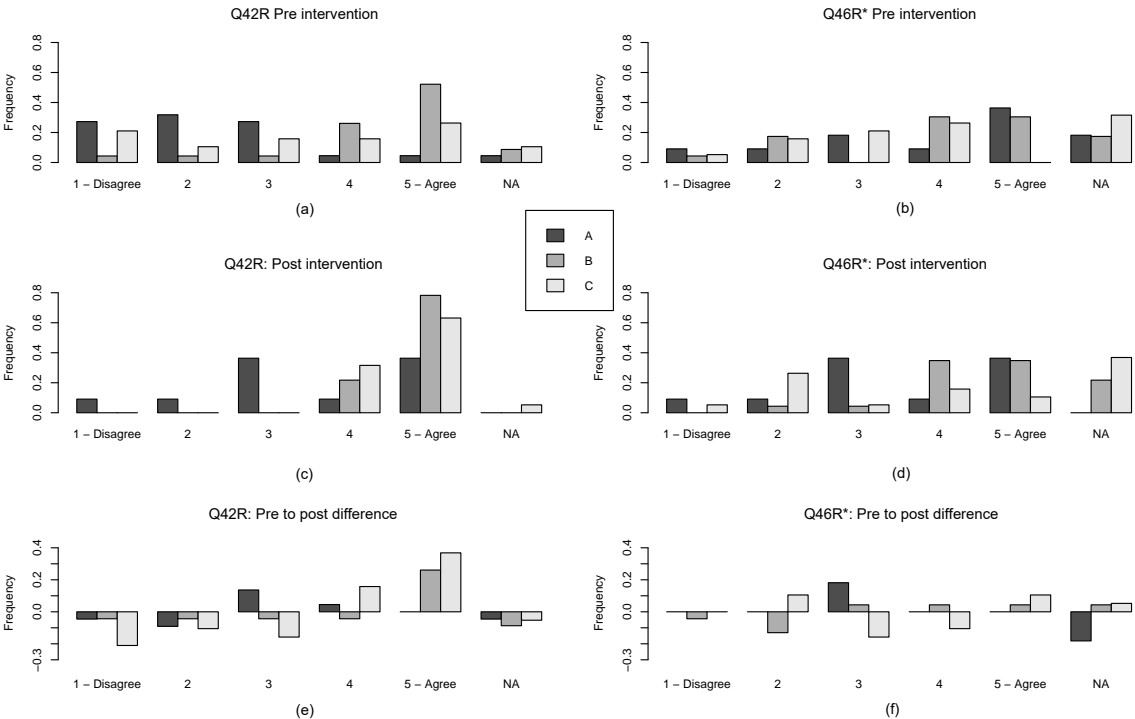

**Figure 7.** Pre intervention and post intervention frequency distributions and differences in these for Super Groups A, B, and C for questions 42R (My teaching will be very effective when using formative assessment) and 46* (Increased effort of the teacher in using formative assessment produces change in some students' achievement in inquiry-based competencies). R means that coding was in reverse, and * marks an outcomes expectation question.

## 5. Discussion

The purpose of this study was to find out how when working with formative assessment strategies in collaboration with researchers and other teachers', individual self-efficacy attributes of practicing teachers are differentially 'affected' in different educational contexts, here represented by Local Working Groups. A network-based methodology to analyse similarity of teachers' responses to attributes of self-efficacy questions was developed to meet this purpose. The results showed that based on similarity of answers, there were heterogeneities among groups.

For example, the methodologically identified Super Group A did not perceive themselves to have the necessary skills to use formative assessment strategies in their teaching (Q49R), even if they did know the necessary steps to teach effectively using formative assessment (Q40). We see this as a gap between knowledge and practice, which, if verified, would be useful during teacher development. The patterns of the Super groups can alert teacher educators to unique strengths and challenges of teachers in their local and national context, providing more finely tuned efforts to affect self-efficacy attributes with Bandura's suggestions for change [10]. At the same time, Super Group A showed a lot of diversity in their responses with regards to more specific aspects of the necessary skills. On the other hand, Super Group B responses were similar and were indicative of high self-efficacy attributes. Super Group B did show some variation on outcome expectations, which may be indicative of some uncertainty in the group about the possible effects of formative assessment on student learning. Super Group C's responses represented slightly lower self-efficacy attributes, their answers were not as diverse as Super Group A's but more diverse than Super Group B's. Particularly, their responses to outcome expectation showed the most diversity (entropy), possibly signifying more uncertainty about the effects of formative assessment on student learning than was found in Super Group B.

With the approach described here, new detailed patterns, derived from the raw data, become visible. The pre intervention to post intervention differences in Figures 6 and 7 show differential changes for each of the Super groups, which offer useful patterns of responses to self-efficacy attributes. Our analyses provided illustrative examples of such patterns in comparing shifts from pre intervention to post intervention on responses to questions Q40, Q49, Q42, and Q46. While not conclusive, these are analyses of changes and suggest how learning processes could be described differently for different groups of participating teachers.

The network analyses for Super Groups A, B and C (Figure 3a,b) showed cluster relationships based on similarity of post intervention answers to the self-efficacy questions. They showed that Super Group A was composed largely by teachers from LWG 2. Super Groups B and C were more heterogeneous with regards to LWGs. This study's analysis does not reveal why the different patterns between super groups in self-efficacy attributes occur. The analysis of over-representation showed no evidence of teaching subject or years of teaching experience being factors. Instead, LWG origins did show evidence of over representation. However, any influence from LWG on self-efficacy development remains speculative. The question of why, might in principle be answered quantitatively if (1) the data consisted of a larger group and/or (2) other variables, for example, more detailed knowledge about participant teaching practices and school context, were integrated into the analyses. Likewise, a detailed qualitative analysis of teacher-teacher and researcher-teacher interactions in the six LWGs could have been coupled with the analyses of this paper to produce explanations as to change patterns self-efficacy attributes. Still, the results of this methodology raise interesting questions that may be posed to data resulting from this kind of study.

One set of questions pertains to the details of the distributions of answers to the questionnaire. Does Super Group A's apparent agreement on a lack of skills and their diversity in terms of which skills they seem to lack, mean that a further targeted intervention might have raised their self-efficacies? For instance, the intervention might have used the Post Groups as a basis for coupling teachers with similar problems. Does the diversity of Super Group C on outcome expectations mean that their experiences of how formative assessment affected students does not match with literature suggesting that most or all students benefit from formative assessment strategies? We need further data from the LWGs to develop and confirm these and other newly emerging hypotheses. The network analyses have given us more detailed and insightful looks into the self-efficacy attributes of our project participants than traditional pre intervention to post intervention overall comparison would provide.

Another set of questions pertains to background variables and point to a finer grained analysis of activities in the LWGs. Previous analyses [27] showed mainly positive, yet somewhat mixed changes on each question for the whole cohort. We argue that the network analyses presented here have yielded greater understanding via our characterisation of the super groups. Using the background variable descriptions of super groups (e.g., that LWGs 2 and 4 were over-represented in Super Group A but that subject and years of teaching were not) may help identify on which LWGs to focus. For example, records of LWG activity during the intervention might be used to find opportunities for changes in self-efficacy attributes that may have shaped responses as found in the super groups. Such insights may provide a finer grained understanding of the changes in teacher confidence using formative assessment methods of the project.

A third set of questions pertains to how this kind of analysis could be used in future projects. While we have made the case that this yields a finer grained analysis and thus deeper understanding, we also believe that there is room for integrating the analysis into an earlier stage in research. Working with LWGs means that researchers not only act as observers but purposefully interact with teachers to help them implement research-based methods for teaching, for example, formative assessment methods. Researchers are also purposefully providing self-efficacy raising experiences for the teachers. Knowing how different teachers manage different kinds of interventions may help researchers develop and test training methodologies in the future. In using this method in future interventions studies we propose that researchers might use the Post Groups as an even more

detailed unit of analysis. Each Super Group consisted of 3–5 Post Groups each with 3–9 participating teachers. Knowing each of these groups response patterns might help researchers design specific interventions for teachers represented in a Post Group. For instance, if a group shows the same gap between knowledge and skill as Super Group A, researchers may provide opportunities for guidance. For example, researchers could help with a detailed plan for employing a particular strategy of formative assessment, then observe teaching and facilitate reflection afterwards for these teachers. Such a strategy is costly to employ on a whole cohort level, but using the described methodology would allow researchers to focus on particular groups.

Such a strategy would benefit from more measurements during the project. With more measurements, for example after a targeted intervention as just described, researchers might be able to gauge to what extent teachers' self-efficacy attributes changed. Such a strategy would also better reflect the nature of self-efficacy as malleable and constantly changing.

Given the organization of teachers into LWGs, a fourth set of questions pertains to the group dynamics of each LWG. The literature offers several theoretical frameworks for analyses of group dynamics, for example, Teacher Learning Communities (TLC) [46,47] or Communities of Practice [48] (CoP). A TLC requires "formation of group identity; ability to encompass diverse views; ability to see their own learning as a way to enhance student learning; willingness to assume responsibility for colleagues' growth" ([46], p. 140). A CoP in the sense of Wenger [48] requires a joint enterprise, mutual engagement and a shared repertoire. While a deeper analysis of e.g., LWG meeting minutes in a TLC- or CoP-perspective might help explain some of the findings of this study , this analysis in itself was not designed to capture elements of neither TLCs nor CoPs.

One more result merits a discussion of a future research direction. We found that the pre intervention groups were different from the post intervention groups (Post Group 1–12). However, network analysis offers tools with which to analyse differences in network group structure [33]. One method is a so-called alluvial diagram [49], which visualises changes in group structure. Using such tools, one could investigate if teachers move groups together, which responses remain constant and which change for such moving groups. However, with our current data set, any explanation of such potential changes would be highly speculative.

Obviously, the study is limited because of the size of the population of teachers. Thus, it is not certain that a new study will produce Super Groups that resemble the groups found in this study. A second issue is that we have not verified the validity of the self-efficacy attributes of teachers in each group. Even though self-efficacy attributes are commonly assessed with questionnaires, the added dimensions this network analysis reveals may be validated through, for example, teacher interviews, meeting minutes, and observations. The major strength of the methodology seems to us to be that it allows us to scrutinize the data in new ways and ask questions that we find relevant and which we were not able to ask before.

## 6. Conclusions

This article has presented a way of viewing changes in the use of formative assessment methods in inquiry-based science education through changes in participating teacher self-efficacy attributes. This was illustrated with Local Working Groups, where teachers and researchers discussed and developed actual implementations of different formats for formative assessment.

To analyse the emergent complex data, this study developed and applied a novel methodology for analysing questionnaire data, which relies on similarity of participant answers and network analysis.

Using this methodology, we were able to characterise super groups, yielding a fine-grained understanding of how self-efficacy attributes develop for different groups. As an example, the analysis revealed that members of Super Group A often believed that they were not well prepared to use these formative assessment methods effectively, but during the course of the project they acquired knowledge of the methods, but not high confidence in being able to implement them. In contrast, Super Group B apparently responded to the introduction and use of formative assessment methods in

a positive way by increasing self confidence in their knowledge and use of the methods introduced in the program.

These examples highlight what we believe is the main contribution of this article: This fine grained method of analysis has the potential to be used in other projects or in teaching to pinpoint groups, which seem to respond differently to interventions and modify guidance or instruction accordingly.

Given the novelty of the methodology further research is needed to explain why these differential changes occurred.

**Supplementary Materials:** The following are available online at http://www.mdpi.com/2227-7102/10/3/54/s1: S1. Calculating similarity, S2. Creating a similarity network, S3. Sparsification of network, S4. Community detection for finding groups. S5. Technical results. S6. Finding and characterising the optimal super group solution, S7. Graphs of pre intervention to post intervention, S8. References. Data and R-scripts are available at http://bit.ly/bruunEvans2020.

**Author Contributions:** Conceptualization, J.B. and R.H.E.; Data curation, J.B. and R.H.E.; Formal analysis, J.B. and R.H.E.; Funding acquisition, R.H.E.; Investigation, R.H.E.; Methodology, J.B.; Visualization, J.B. and R.H.E.; Writing—original draft, J.B. and R.H.E. All authors have read and agree to the published version of the manuscript.

**Funding:** This research was supported by European Commission [grant number FP7 Grant Agreement No. 321428 ASSIST-ME].

**Conflicts of Interest:** The authors declare no conflict of interest. The funders had no role in the design of the study; in the collection, analyses, or interpretation of data; in the writing of the manuscript, or in the decision to publish the results.

## Abbreviations

The following abbreviations are used in this manuscript:

| | |
|---|---|
| CoP | Communities of Practice |
| LWG | Local Working Group |
| LANS | Locally adaptive network sparsification |
| NMI | Normalized Mutual Information |
| TLC | Teacher Learning Communities |
| Q | Modularity |

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
