# Peer review of "Network Analysis of Survey Data to Identify Non-Homogeneous Teacher Self-Efficacy Development in Using Formative Assessment Strategies"

_education, doi:10.3390/educsci10030054_

Round 1

Reviewer 1 Report

This is a highly complex paper and might be interesting to those who work with network analyses. I must confess that I am not sure at every point to have understood why certain decisions during the analysis were done, caused by the method or just decisions of the authors. Unfortunately, the findings on the supergroups do not yet found explanations. This is how research sometimes is.

The paper uses LWG. Is this different form Teacher Learning Communities or Communities of Practice. It is not really clear what happened in the meetings, this might need a bit more information. It also needs to be said if the project understood itself as some top-down training, or a bottom-up learning community – and whether this interpretation was coherent in all LWGs?

Section 3 needs checking. There are many information missing that pop up later in the paper, e.g. that we work with Likert scales and that they are 5-step. You need to clarify early:

Section 3.1. (a) in the final MS the names of the countries should be given (b) teaching experiences should be quantified (what is few, several and many?) Section 3.2. How often die the LWGs met, in what frequency and over what amount of time? 3. is confusing or not sufficiently explained. You take 12 out of 23 and have Q38 and Q43? It becomes clear by Figure 2, but needs to be clear already in the text. Format is unclear, where these Likert items? How many steps? Later it says teachers took option 2, what does this mean. It needs to be clear from the text at this point without searching in the later text or supplemented files.

Check the whole MS for this issue.

The authors may read the MS again, there are a couple of chance for improvement, e.g. whether to use hyphens when talking about pre and post, or not.

Author Response

We thank the reviewer for taking time to review our paper and providing comments that we could use to improve upon the manuscript. We have made extensive changes to the manuscript based on both reviewer comments. One of the more substantial changes is our decision to remove all analyses of aggregated scores. Instead, we have included clarifications along the lines of the suggestions from the reviewers and we have focused on the single-item nature of our analyses. Here, we detail our work with focus on Reviewer 1's comments. Given the many changes, we have also uploaded a pdf-file that tracks all changes made.

Reviewer point 1: This is a highly complex paper and might be interesting to those who work with network analyses. I must confess that I am not sure at every point to have understood why certain decisions during the analysis were done, caused by the method or just decisions of the authors. Unfortunately, the findings on the supergroups do not yet found explanations. This is how research sometimes is.

Our response: We agree that the paper is complex and we hope that the reviewer with our extensive changes may see that our analyses could be beneficial for developmental work as well as of interest for people interested in network analyses.

Reviewer point 2: The paper uses LWG. Is this different form Teacher Learning Communities or Communities of Practice. It is not really clear what happened in the meetings, this might need a bit more information. It also needs to be said if the project understood itself as some top-down training, or a bottom-up learning community – and whether this interpretation was coherent in all LWGs?

Our responses: We have done several things to address these issues. First, we have added a paragraph explaining briefly, what happened at the meetings (lines 77-84). Furthermore, in our methods section, we have made changes to clarify the process of the intervention. We have not used the terms top down or bottom up, but in lines 77-88, we believe that our introduction could be labelled top-down. However, discussions of experiences and further use could be considered bottom-up. However, we did not use these terms in any stage of our process. We agree that it is very interesting, whether interpretations of the process and the formative assessment methods were consistent across LWGs. However, we believe that this would require extensive analyses of e.g. meeting minutes with associated different frameworks and research questions. We have addressed this issue in lines 560-568 in the Discussion.

Reviewer point 2: Section 3 needs checking. There are many information missing that pop up later in the paper, e.g. that we work with Likert scales and that they are 5-step. You need to clarify early:

Section 3.1. (a) in the final MS the names of the countries should be given (b) teaching experiences should be quantified (what is few, several and many?) Section 3.2. How often die the LWGs met, in what frequency and over what amount of time? 3. is confusing or not sufficiently explained. You take 12 out of 23 and have Q38 and Q43? It becomes clear by Figure 2, but needs to be clear already in the text. Format is unclear, where these Likert items? How many steps? Later it says teachers took option 2, what does this mean. It needs to be clear from the text at this point without searching in the later text or supplemented files.

Check the whole MS for this issue.

Our responses: (3.1.a) We argue that since LWGs cannot be said to be representative of the countries in which they were embedded, we cannot make any meaningful interpretations based on LWG country. We therefore opt not to include this information. We address this point in lines 194-196.

(3.1.b) We have corrected this to provide the original categories (less than 4, 5-10, 11-20, and more than 20). See lines 166-167.

(3.2) We have added information about the project's three half-year 'trials' as well as meeting frequencies (3-4 times/trial) to section 3.2, lines 173-180.

(3.3) We have now added information in lines 198-215. Here, we clarify that the 12 SE-attribute items were part of a larger questionnaire for the teachers in the project. We also clarify in the text that these were likert-items with 5 steps with a sixts option of 'no response'. In line 233 we have added "likert" so that it is clear that the teacher is choosing likert option '2'. We have checked the manuscript to be consistent with this terminology.

Reviewer comment 4: The authors may read the MS again, there are a couple of chance for improvement, e.g. whether to use hyphens when talking about pre and post, or not.

Our response: We have re-read the MS and have settled on the phrasing "pre intervention" and "post intervention". 

Reviewer 2 Report

The paper aims to introduce a novel method to analyze how teachers' self-efficacy develops over the course of a four year intervention on a finer grain size than what a simple pre/post analysis would reveal. 

I think in general, the authors achieve this goal in a overall well written paper. However, I have two major concerns with the paper in its present state.

At several points during the paper the authors make an argument for their network analytical method being better than other already existing methods, for example they they say that PCA or FA is not feasible with the small sample (L 208) and that network analysis also allows to describe how each person fits into a detected group and how similar the intensified groups are to each other (130). Later in the paper the authors contrast their analysis with some more traditional analyses and say that the traditional analysis does not fully recover the network results. To me, these arguments are not convincing. I don't see how mixture models, cluster analysis, FA with regularization to account for the small N and other models could not provide similar results. Further, the researcher degrees of freedom make it impossible to rule out slightly different specifications of he tertile analysis would not yield results identical to those that the networks provided. This is not to say that I disagree with the network analysis per se. I just do not see the need to demonstrate how the method is superior, especially given that the authors do not demonstrate this comprehensively and this is not a methods journal. I would rather be interested in a more detailed discussion of the results and their possible implications.

My second major point considers that the authors never discuss that they effectively analyze single items. The original instrument was, to the best of my knowledge, developed as a scale for self efficacy and validated in two studies. These studies apparently suggest a one factor solution and high reliability for the scale. The factor loadings also suggest that different individual question do not solely measure self-efficacy. What measures self-efficacy is the predicted score from the factor model. Thus, the question arises what you are actually measuring when you look at individual questions, still self-efficacy, a different construct, or a mixture thereof. Further, what about the reliability of these single item? I think this issue needs to be discussed, rather than the potential superiority of the network measure of traditional measures.

In the following I list some minor issues.

Generally, I would like to see effect sizes and degrees of freedom reported with statistical analysis. Degrees of freedom are important to be able to judge the soundness of the analysis and effect sizes help the interpretation of the results.

In the theory section, the authors write that "self-efficacy beliefs are malleable and constantly changing" (L 55). If this really is the case, why would one care about a two point measurement of this construct? Wouldn't this require many more measurement points and respective time-series models? Also, in this light, the overall rather small differences from pre to post and between the groups (Fig 4) appear hard to interpret? Couldn't this just be random if self efficacy is constantly changing?

The authors say that they found the results of the network analysis were not stable between pre and post. In consequence they focus their analysis on the post-networks since they are interested in the effect of the intervention. However, wouldn't the effect of the intervention be the change between pre and post? Thus, wouldn't changes in network structure and changes in group membership be potentially meaningful? A potential way to look at this would be transition models or a multiplex network of pre and post. 

Similarly, the analysis often compares post scores of the supergroups. I would rather be interested in differences of change between pre and post for the supergroups. I think this is what better reflects actual learning.

Figure S3 is missing in supplementary materials.

Figure 4, what do the graphs show? Averagers and 95% confidence intervals?

In sum, I generally think that looking at data in a more nuanced why is a good idea and networks analysis offers an interesting way to do this but the paper does not conveince me that networks analysis is superior and there are some methodological choices that need to be better explained / discussed / justified.

Author Response

We wish to thank the reviewer for taking time to read and provide useful and insightful comments to our manuscript. We have used both reviewer comments to make extensive changes to the manuscript. We outline these changes in the cover letter. Below, we provide detailed responses to Reviewer 2:

Reviewer comment 1: At several points during the paper the authors make an argument for their network analytical method being better than other already existing methods, for example they they say that PCA or FA is not feasible with the small sample (L 208) and that network analysis also allows to describe how each person fits into a detected group and how similar the intensified groups are to each other (130). Later in the paper the authors contrast their analysis with some more traditional analyses and say that the traditional analysis does not fully recover the network results. To me, these arguments are not convincing. I don't see how mixture models, cluster analysis, FA with regularization to account for the small N and other models could not provide similar results. Further, the researcher degrees of freedom make it impossible to rule out slightly different specifications of he tertile analysis would not yield results identical to those that the networks provided. This is not to say that I disagree with the network analysis per se. I just do not see the need to demonstrate how the method is superior, especially given that the authors do not demonstrate this comprehensively and this is not a methods journal. I would rather be interested in a more detailed discussion of the results and their possible implications.

Our responses: Upon scrutinizing the results, we agree with the reviewer that we did not demonstrate a comprehensive superiority of the method. Thus, we have removed these parts of the manuscript. We have retained a discussion about this in lines 137-143, which we have modified, so we no longer claim that network analysis is superior but does have other qualities than alternative methods. 

We have provided a more detailed dicussion of the results and possible implications (see below).

Reviewer comment 2: My second major point considers that the authors never discuss that they effectively analyze single items. The original instrument was, to the best of my knowledge, developed as a scale for self efficacy and validated in two studies. These studies apparently suggest a one factor solution and high reliability for the scale. The factor loadings also suggest that different individual question do not solely measure self-efficacy. What measures self-efficacy is the predicted score from the factor model. Thus, the question arises what you are actually measuring when you look at individual questions, still self-efficacy, a different construct, or a mixture thereof. Further, what about the reliability of these single item? I think this issue needs to be discussed, rather than the potential superiority of the network measure of traditional measures.

Our responses: We agree with the reviewer that we analyse single items and that we cannot claim that we are measuring general self-efficacy constructs. This is part of our reasoning behind leaving out all analyses pertaining to aggregated SE-scores. In stead, we make clear that we are addressing self-efficacy attributes. In lines 96-113 and lines 206-215 we address the issue to make clear that we analyse single items. While it would be interesting to compare the single-attribute view with the aggregated view to see what is measured with the 12 items, we do not believe this is possible when we do not use the whole STEBI-B instrument.

Reviewer comment 3: Generally, I would like to see effect sizes and degrees of freedom reported with statistical analysis. Degrees of freedom are important to be able to judge the soundness of the analysis and effect sizes help the interpretation of the results.

Our responses: We have included degrees of freedom in all our

Reviewer comment 4: In the theory section, the authors write that "self-efficacy beliefs are malleable and constantly changing" (L 55). If this really is the case, why would one care about a two point measurement of this construct? Wouldn't this require many more measurement points and respective time-series models? Also, in this light, the overall rather small differences from pre to post and between the groups (Fig 4) appear hard to interpret? Couldn't this just be random if self efficacy is constantly changing?

Our responses: We absolutely agree with the reviewer on this crucial point. This was part of our decision to remove the pre-to-post score analyses from the paper. Thus, the original Figure 4 is no longer part of the manuscript as is true for pre-to-post intervention analyses of accumulated scores. However, as we show, current research rely on pre intervention to post intervention change designs, which is why we believe that analysing pre intervention to post intervention data is still relevant.

What we hope to show now is that more than two points of observation may be beneficial when working to change teacher self-efficacy. See our discussion of this in lines 547-560.

Reviewer comment 5: The authors say that they found the results of the network analysis were not stable between pre and post. In consequence they focus their analysis on the post-networks since they are interested in the effect of the intervention. However, wouldn't the effect of the intervention be the change between pre and post? Thus, wouldn't changes in network structure and changes in group membership be potentially meaningful? A potential way to look at this would be transition models or a multiplex network of pre and post. 

Our responses: We agree with the reviewer that transition models would portray the effects of the intervention. In lines 570-576 we discuss this and provide a possible future way of doing this. However, without analyses of other data relevant to describing the intervention dynamics (such as meeting minutes), we believe that possible explanations would be speculative at this point.

Reviewer comment 6: Similarly, the analysis often compares post scores of the supergroups. I would rather be interested in differences of change between pre and post for the supergroups. I think this is what better reflects actual learning.

Our responses: We agree with the reviewer that changes between pre intervention and post intervention are much more interesting than scores. We have provided two illustrative examples of this. Se lines XXX_YYY. However, we do not provide an exhaustive analysis of all pre intervention to post intervention analyses. This is because we do not analyse other data (such as meeting minutes) that might help provide explanations at this level of detail. Instead, we have opted to show two illustrative examples. See pages 14-15 and Figures 6 and 7. We have included figures that show pre intervention, post intervention and shifts for all questions in the Supplemental File.

Reviewer comment 7: Figure S3 is missing in supplementary materials.

Our response: Figure S3 is now included in a revised form.

Reviewer comment 8: Figure 4, what do the graphs show? Averagers and 95% confidence intervals?

Our response: Figure 4 has been removed due to the changed focus of the paper.

Reviewer comment 9: In sum, I generally think that looking at data in a more nuanced why is a good idea and networks analysis offers an interesting way to do this but the paper does not conveince me that networks analysis is superior and there are some methodological choices that need to be better explained / discussed / justified.

Our response: In addressing both reviewer comments in detail, we believe that our paper now better shows how network analyses can be used fruitfully rather than having a misplaced focus on the superiority of the method.

Round 2

Reviewer 2 Report

Dear authors, 

I appreciate how thoroughly you responded to the comments and resolved the issues.

I would recommend that you align the way you report statistics of t-tests etc with APA style which is predominantly used in education research.

Author Response

Reviewer comment 1: I appreciate how thoroughly you responded to the comments and resolved the issues.

Our response: Thank you!

Reviewer comment2: I would recommend that you align the way you report statistics of t-tests etc with APA style which is predominantly used in education research

Our response: We have gone through the document and aligned our reports with APA standards. For the Kruskall-Wallis tests, we we able to find only this web-site which suggests a reporting format similar to APA. For the post hoc tests Nemenyi tests we were unable to find any APA style format, and chose to keep this. Z tests were to the best of our knowledge already aligned with APA style, and we have also aligned means and standard deviations to APA style.